# Autophagy and oxidative stress modulation mediate Bortezomib resistance in prostate cancer

Kalliopi Zafeiropoulou[1,2], Georgios Kalampounias[1], Spyridon Alexis[2], Daniil Anastasopoulos[1], Argiris Symeonidis[2], Panagiotis Katsoris[1]*

1 Division of Genetics, Cell Biology and Development, Department of Biology, University of Patras, Patras, Greece, 2 Hematology Division, Department of Internal Medicine, University of Patras Medical School-University Hospital, Patras, Greece

☯ These authors contributed equally to this work.

* katsopan@upatras.gr

## Abstract

Proteasome inhibitors such as Bortezomib represent an established type of targeted treatment for several types of hematological malignancies, including multiple myeloma, Waldenstrom's macroglobulinemia, and mantle cell lymphoma, based on the cancer cell's susceptibility to impairment of the proteasome-ubiquitin system. However, a major problem limiting their efficacy is the emergence of resistance. Their application to solid tumors is currently being studied, while simultaneously, a wide spectrum of hematological cancers, such as Myelodysplastic Syndromes show minimal or no response to Bortezomib treatment. In this study, we utilize the prostate cancer cell line DU-145 to establish a model of Bortezomib resistance, studying the underlying mechanisms. Evaluating the resulting resistant cell line, we observed restoration of proteasome chymotrypsin-like activity, regardless of drug presence, an induction of pro-survival pathways, and the substitution of the Ubiquitin-Proteasome System role in proteostasis by induction of autophagy. Finally, an estimation of the oxidative condition of the cells indicated that the resistant clones reduce the generation of reactive oxygen species induced by Bortezomib to levels even lower than those induced in non-resistant cells. Our findings highlight the role of autophagy and oxidative stress regulation in Bortezomib resistance and elucidate key proteins of signaling pathways as potential pharmaceutical targets, which could increase the efficiency of proteasome-targeting therapies, thus expanding the group of molecular targets for neoplastic disorders.

## Introduction

The ubiquitin–proteasome pathway is the most important intracellular proteolytic system, and the integrity of its function is crucial for cell homeostasis [1]. Proteasome substrates encompass signaling molecules, tumor suppressors, cell-cycle regulators, transcription factors, inhibitory molecules (whose degradation activates other proteins), and anti-apoptotic proteins (e.g., Bcl-2), among others [2]. When degradation of these proteins is blocked, the detrimental effect

**Funding:** The authors received no specific funding for this work.

**Competing interests:** The authors have declared that no competing interests exist.

is potentially enormous, especially for rapidly dividing cancer cells, which require increased availability of growth-promoting proteins to sustain the accelerated and uncontrolled rate of mitosis characteristic of cancer cell development and spread [3]. Consequently, inhibition of the proteasome may delay cancer progression by interfering with the regular degradation of cell-cycle proteins. Indeed, the inhibition of proteasomal function results in the induction of programmed cell death in several cell lines (apoptosis) [4–7], and therefore, proteasome inhibitors have been characterized as potential anticancer drugs.

Bortezomib, initially designated as PS-341 (Velcade®), reversibly inhibits the chymotrypsin-like activity of the proteasome. Chemically, Bortezomib is a dipeptidyl boronic acid analog derived from leucine and phenylalanine. It has been shown to inhibit tumor cell proliferation, adhesion, and metastasis in many in vivo and in vitro models and has been approved by the US Food and Drug Administration (FDA) for use in cancer treatment [8]. Several clinical trials have revealed that Bortezomib can be used to treat many types of solid tumors alone or in combination with other chemotherapeutic drugs. Bortezomib, first approved for patients with multiple myeloma, has also demonstrated inhibitory effects on colon-gastric cancer [9, 10], breast cancer [11, 12], prostate cancer [13, 14], and lung cancer [15, 16] as well as other cancer types.

Despite the impressive initial response, Bortezomib efficacy is limited by the rapid emergence of resistance. Almost 20 years after its approval, the mechanism of resistance to Bortezomib remains unclear. Several studies have been conducted to fully understand the mechanism underlying this resistance. Recently, some of the mechanisms of drug resistance have been elucidated. In vitro studies in lymphoma and leukemia cell lines have shown that Bortezomib resistance is developed either following mutations of the β5-subunit gene and/or β5 proteasome subunit gene overexpression [17, 18] that leads to elevated chymotrypsin-like activity [19].

Normally, Bortezomib stabilizes p21, p27, and p53, as well as some pro-apoptotic proteins [20, 21] leading to tumor cell death. On the contrary, Bortezomib-resistant cells have been proven to evade apoptosis by losing their ability to stabilize and accumulate pro-apoptotic proteins [22]. However, despite the importance of cell cycle regulation in cancer, there is little information about the precise role of cell cycle regulators in the development of Bortezomib resistance. Additionally, resistance has been associated with elevated phosphorylation levels of pro-survival proteins belonging to MAPKs, PIP3/AKT/mTOR, and JAK/STAT pathways, as well as of the underlying crosstalk [11, 23, 24]. Key molecules of these pathways have been reported to be activated inside the resistant cells, elucidating the importance of survival protein modulation [25]. Furthermore, upregulation of pathways that suppress apoptosis and induce autophagy [26, 27] has also been reported in bortezomib-resistant cells. It is well known that these two mechanisms' interplay is crucial for cell survival. Autophagy blocks the activation of apoptosis, and in turn, apoptosis blocks the activity of autophagy through caspase-mediated cleavage of the autophagic proteins [28]. Autophagy refers to a set of pathways by which abnormal, malfunctioning, or simply excessive cytoplasmic material is delivered into the lysosomes for degradation [29]. Autophagy is linked to the UPS [30], regulating proteostasis as well as partially substituting it [31, 32]. Many multidrug-resistant types of cancer have been reported to possess upregulated autophagy biomarkers [33]. In parallel, p62 and LC3 are elevated, and an important role has been assigned to Beclin-1, which regulates the autophagy-apoptosis equilibrium [34].

Some recent reports indicate ROS-induced cell cycle arrest as a mechanism of drug resistance [35]. However, until today, there is no clear evidence that ROS levels are crucial for the establishment of Bortezomib resistance, although modulating intracellular ROS levels appears to be crucial for overcoming multidrug resistance in cancer cells [36]. ROS levels may affect

the phosphorylation of these cell cycle regulators and, influence cell cycle progression. Thus, G1-arrested melanoma cells were resistant to apoptosis induced by the proteasome inhibitor bortezomib, irrespective of the factor mediating the arrest, a finding suggesting that induction of G1 arrest may result in Bortezomib resistance [37].

Prostate cancer is a very common type of malignancy emerging ιn men, for which Bortezomib has not been approved yet. Even though prostate cancer cells exhibit an initial susceptibility to the drug, the development of resistance remains a major problem, limiting the use of Bortezomib. Clinical trials in this field are limited [38], because of a lack of data adequately investigating the resistance mechanisms even in in-vitro conditions. In the present study, we established a Bortezomib-resistant prostate cancer cell line, DU-145 RB60, to study the differential effects of the drug in naïve (DU-145) and bortezomib-resistant (DU-145 RB60) cells. Hereby, we focused on changes in the accumulation of β5 subunits (PSBM5), poly-ubiquitinated proteins, cell cycle, and autophagy regulators, as well as on the apoptotic rate and ROS levels in both cell populations.

## Materials and methods

### Reagents and materials

Cell culture medium RPMI 1640 and all other culture reagents were purchased from Biowest (France). The specific proteasome inhibitor, Bortezomib (formerly known as Velcade$^{TM}$), was generously provided by Janssen-Cilag (Greece). All culture plates were purchased from Greiner Bio-One (Austria).

### Cell culture

The human prostate cancer epithelial cell line DU-145 (ATCC) was cultured in RPMI 1640 medium, supplemented with 10% Fetal Bovine Serum (FBS), 100 units/ml penicillin, and 100 μg/ml streptomycin. Cultures were incubated at 5% $CO_2$ and 100% humidity at 37˚C.

### Cell viability assay

To assess viability, cells were plated at a density of 50.000 cells per well inside 24-well plates. After cell spreading and adhesion, they were treated with the indicated concentrations of Bortezomib, Carfilzomib or Doxorubicin in RPMI 1640 supplemented with 10% FBS for 24 to 72 h. The inhibitory effect of each drug on cell growth was measured using the crystal violet assay [39]. Adherent cells were fixed with methanol and stained with a 0.5% crystal violet in 25% methanol aqueous solution for 20 min. After gentle rinsing with water, the retained dye was extracted using a 30% acetic acid aqueous solution, and the absorbance was measured at 595 nm using a plate reader spectrophotometer. The $IC_{50}$ were estimated using built in equations provided with GraphPad Prism 8.

### Microchemotaxis/Transwell chambers

Known numbers of cells were transferred inside Transwell chambers/inserts containing serum-free medium with or without Bortezomib. The inserts were then submerged inside wells of 24-well plates containing medium supplemented with 20% FBS with or without Bortezomib. The cells were left to migrate for an indicated interval and then were washed with PBS solution and fixed with methanal solution. The fixed cells were stained with 0.33% toluidine blue solution, and photographs were taken on a photonic microscope at x20 magnification [39]. The cells in each photograph were counted using the Cell Counter ImageJ built-in tool, and the results were analyzed using Graphpad Prism 8.

## Wound healing assay

Cells were grown on 6-well plates until they formed a confluent monolayer. Subsequently, cells were scratched in a cross-like manner using a tip, and the medium was replaced with Bortezomib-containing RPMI 1640 supplemented with 10% FBS. Photographs were taken immediately after the scratches/wound formation as well as after key time points [40]. The wound closure percentage was calculated using the ImageJ Manual Wound Healing Size tool [41], and the results were analyzed with GraphPad Prism 8.

## Proteasome activity assay

Cells were exposed to a range of various Bortezomib concentrations for 24 h. Total proteins were extracted from the cells using sonication and incubation in a solution containing 50 mM HEPES, 20 mM KCl, 5 mM $MgCl_2.H_2O$, and 1 mM Dithiothreitol (DTT) (Cat. 10197777001, Merck) with a pH value of 7.81. The extract was incubated with proteasome fluorogenic substrate-peptide LLVY-AMC (Suc-Leu-Leu-Val-Tyr-7-amide-4-methylcoumarin) (Cat. Number 3120-v, Peptide Institute Inc.), with or without the proteasome inhibitor MG-132 (Cat. Number 3175-v, Peptide Institute Inc.) inside a flat-bottom black polystyrene 96-well plate for fluorescence for 1 h at 37˚C [42]. Fluorescence intensity was measured at 380 nm excitation wavelength and 460 nm emission wavelength. Total protein concentration was estimated using the Q5000 Nanodrop Quawell spectrophotometer.

## Western blot analysis

Cells were starved of any drugs/inhibitors for 24 h, and then incubated with the indicated concentrations of Bortezomib for varying times. Subsequently, they were washed twice with a PBS solution and lysed using RIPA buffer. Total proteins were determined using the Bradford assay. Equal amounts of total proteins were mixed with Laemmli's Sample Buffer 2X solution containing 5% β-ME, and the samples were denatured at 95˚C for 10 min. Proteins were separated by SDS-PAGE and transferred to an Immobilon-P membrane (Millipore, USA) for 30 min using Towbin's transfer buffer in a semi-dry transfer system. The membrane was blocked in TBS containing 5% skimmed milk and 0.1% Tween-20 for 1 h at 37˚C. Membranes were then probed with primary antibodies (Table 1) overnight at 4˚C, under continuous agitation.

The blot was then incubated with the appropriate secondary antibodies (Anti-rabbit IgG Antibody, CST#7074, or Anti-mouse IgG antibody, CST# 7076) (both diluted 1:2000) coupled to horseradish peroxidase, and bands were detected with the SuperSignal™ West Femto

**Table 1. List of primary antibodies used for western blot analyses.**

| Antibody | Supplier | Dilution |
|---|---|---|
| b-actin antibody (monoclonal mouse) | CST #3700 | 1: 2000 |
| Proteasome 20S β5 subunit antibody (polyclonal mouse) | Santa Cruz #sc-393931 | 1: 1000 |
| Ubiquitin antibody (P4D1) (monoclonal mouse) | Santa Cruz #sc-8017 | 1:2000 |
| p21 antibody (monoclonal rabbit) | CST #2947 | 1:1000 |
| p27 antibody (monoclonal rabbit) | CST #3686 | 1:1000 |
| p53 antibody (monoclonal mouse) | Invitrogen # MA5-12557 (ThermoFisher Scientific) | 1:500 |
| p44/42 MAPK (Erk1/2) Antibody (polyclonal rabbit) | CST #9102 | 1:1000 |
| Phospho-p44/42 MAPK (Erk1/2) (Thr202/Tyr204) Antibody (polyclonal rabbit) | CST #9101 | 1:1000 |
| Phospho-p38 MAPK (Thr180/Tyr182) antibody (monoclonal rabbit) | CST #9215 | 1:1000 |

Maximum Sensitivity Substrate (Thermo Scientific™ #34096), according to the manufacturer's instructions. Where indicated, blots were stripped in buffer containing 62.5 mM Tris HCl pH 6.8, 2% SDS, and 100 mM 2-mercaptoethanol for 30 min at 50˚C and reprobed with primary antibodies. Quantitative estimation of band size and intensity was performed through analysis of digital images using ImageJ [43], and wherever statistical analysis was needed, it was performed using t-tests with the GraphPad Prism 8 software.

## Immunofluorescence confocal microscopy

Cells were cultured on glass coverslips (MGF-slides, Germany) for 24 h and then incubated with appropriate Bortezomib concentrations. After this interval, they were gently rinsed with PBS and fixed in a 4% paraformaldehyde aqueous solution for 15 min at room temperature. Subsequently, they were rinsed three times with PBS, permeabilized for 15 min in a PBS solution containing 0.1% Triton X-100 and then blocked in a PBS solution containing 5% skimmed milk for 1 h at room temperature. Cells were then incubated for 1 h, at room temperature with the primary antibodies/fluorophores (Table 2).

After rinsing with PBS Tween-20, the coverslips were incubated with secondary antibodies; Donkey anti-Rabbit IgG (H+L) Highly Cross-Adsorbed antibody, Alexa Fluor™ 647 (Invitrogen, #A-31573) or Goat anti-Mouse IgG (H+L) Cross-Adsorbed secondary antibody, Alexa Fluor™ 488 (Invitrogen, #A-11001) (Dilution 1:500) in permeabilization buffer. After rinsing three times in PBS, cells were mounted using Mowiol 4–88Ⓡ (Sigma). To detect autophagy, live cells grown on coverslips were incubated with Lysotracker RED (Invitrogen) for 15 min and then immediately imaged. Labeling was performed using Leica Confocal Imaging System and photographs were taken using the LAS X software. Lysotracker staining of live cells was also used to quantify the acidic protein content using flow cytometry.

## Flow cytometry

Cells were incubated with the indicated concentrations of Bortezomib in RPMI 1640, supplemented with 10% FBS, for appropriate time intervals (24–48 h). Subsequently, they were stained with Annexin-V conjugated with FITC (Invitrogen™ #A13199) for 20 min at room temperature to detect apoptosis [44]. Propidium iodide (PI) (BD Pharmingen™ #556463) was used both as a necrotic marker during the Annexin V assay, and during the cell cycle analysis to quantify DNA content. Staining with Lysotracker RED was optimized for acidic protein content estimation. The cells were stained for 45 min at 37˚C using the concentration suggested for immunofluorescence and subsequently rinsed and analyzed. Finally, the cell-permeant 2',7'-dichlorodihydrofluorescein diacetate ($H_2$DCFDA) (Invitrogen, # D399) was used to estimate Reactive Oxygen Species (ROS) in these cells. The samples were analyzed with a FACS Calibur cytometer (BD, Biosciences). For each sample, 200,000 ungated events were acquired.

**Table 2. List of primary antibodies used for immunocytochemical staining.**

| Antibody | Dilution |
|---|---|
| beta-actin antibody (monoclonal mouse) | 1: 1000 |
| p21 antibody (monoclonal rabbit) | 1:500 |
| p27 antibody (monoclonal rabbit) | 1:500 |
| DAPI | 0.1 μg/ml |

## Results

### Generation of a DU-145 Bortezomib-resistant clone

Bortezomib-resistant cells were established through a gradual increase in the concentration of Bortezomib for at least 24 weeks, from 5 to 60 nM Bortezomib, and then maintained at the same concentration constantly for 24 weeks. The resistant clone was named DU-145 RB60, and it was cultured in parallel with the naïve DU-145 clone. As it was also crucial to clarify if this acquired phenotype is inherent or can be reversed after long-term drug withdrawal, we maintained a subclone of the DU-145 RB60 deprived of Bortezomib for another 24 weeks to determine the stability of this characteristic. Furthermore, we were able to assess the effects of Bortezomib on the cells' proliferation rate and potential changes in the main signaling pathways. These long-term untreated cells were named the DU-145 RB60U cell clone, and the drug withdrawal was facilitated at the same time the dose was fixed at the 60 nM milestone for the DU-145 RB60 clone (24 weeks since the introduction of Bortezomib).

The clones were examined for Bortezomib-induced growth inhibition. Dose-response viability/proliferation curves showed 5-fold higher resistance ($IC_{50}$: 60 nM) in DU-145 RB60 compared to the naïve DU-145 after 72 h of treatment with Bortezomib during the first 24 weeks of gradual-increasing Bortezomib presence (Fig 1A). Following another 24 weeks of constant Bortezomib presence (a dose of 60 nM), the RB60 exhibited a 10-fold higher resistance ($IC_{50}$: 125.5 nM) (Fig 1A). The deprived clone (DU-145 RB60U) was assessed 24 weeks after the withdrawal of Bortezomib, and the calculated $IC_{50}$ was almost identical to the one measured at the time of the initial inhibitor deprivation, indicating the persistence of the characteristic (Table 3). In addition, the clones were assessed for resistance against the second-generation proteasome inhibitor Carfilzomib (Fig 1B). The DU-145 RB60 clone initially exhibited a 4-fold resistance against Carfilzomib, but after 24 months, the level of resistance increased to an 8-fold change. Even though the RB60 clone was being cultured constantly in the presence of Bortezomib (and was able to withstand doses of >200 nM), the cells remained susceptible to high doses of Carfilzomib (Table 3).

Additionally, we investigated whether resistance to Bortezomib (or proteasome inhibitors in general) would also increase the cells' ability to resist other chemotherapeutic agents like anthracyclines (apoptosis inducers). For this purpose, we calculated the $IC_{50}$ values following incubation with doxorubicin for a 24-hour interval. We found that the resistant cells did not have a changed $IC_{50}$ value regarding Doxorubicin. The $IC_{50}$ of all the clones tested ranged around 341 nM, which was also confirmed bibliographically [45].

The baseline biological activities such as migration, chemotactic movement, and wound healing of the resistant clone DU-145 RB60 were also assessed (Fig 1C and 1D). To study the effect of Bortezomib on migration, we incubated naïve DU-145 cells and DU-145 RB60 cells with or without the presence of the drug in micro-chemotaxis chambers (Fig 1C). In the absence of Bortezomib, the DU-145 RB60 cells migrated with a slightly higher rate towards the chemoattractant medium compared to the naïve clone. The presence of 60 nM Bortezomib in the upper compartment did not inhibit the DU-145 RB60 clone's ability to migrate but rather induced their movement towards the chemoattractant medium (3-fold change). The same was not observed in the naïve clone, where Bortezomib reduced cell motility significantly. When we added 20 nM Bortezomib to the chemoattractant medium (lower compartment), naïve DU-145 cells exhibited reduced motility by ~60%, while the resistant clone decreased its motility only by ~38%, suggesting that the drug acts as a chemorepellent for the naïve cells while the effect is of less significance for the resistant clone. Further dose escalation (to both migration and chemotaxis assays) revealed that doses up to 240 nM were required to achieve the same effects on the resistant clone. Finally, we assessed the wound-healing ability of both clones,

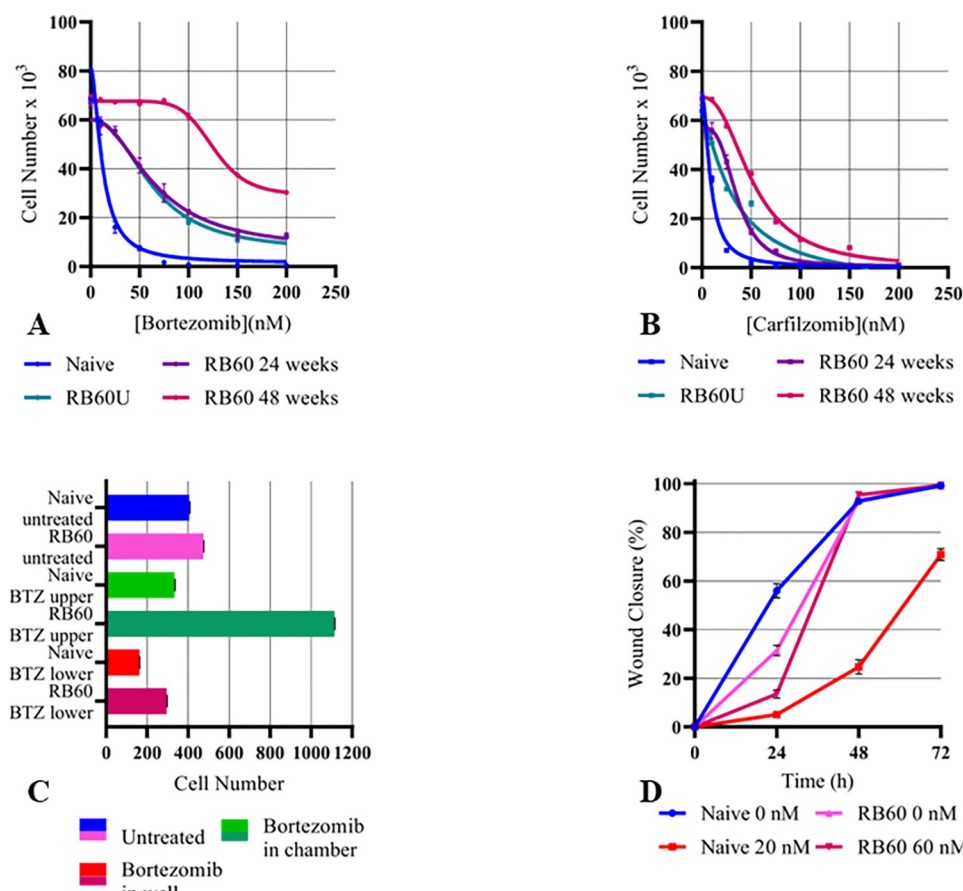

**Fig 1. Main cell function assays of naïve DU-145, DU-145 RB60, and DU-145 RB60U cells.** (**A, B**) Equal numbers of cells were seeded on 24-well plates, and after 24 h of attachment, various doses of Bortezomib (**A**) or Carfilzomib (**B**) were added. Following 72 h of incubation, the cells were fixed and subsequently stained with crystal violet. The proliferation rate of naïve DU-145, DU-145 RB60 resistant cells after 24 weeks of drug presence, DU-145 RB60U cells after 24 weeks of drug absence, and DU-145 RB60 cells after 48 weeks of drug presence (60 nM) was assessed by a spectrophotometrical determination of the crystal violet solution's O.D. at 595 nm. The analysis was performed using the built-in tools provided with the GraphPad Prism 8 software. (**C**) Cells were transferred into a chamber containing serum-free medium with or without Bortezomib. The chambers were placed inside microplates' wells containing medium supplemented with 20% FBS and left to migrate for 24 h. The cells crossing the porous filter were then fixed and stained, and after photographing, they were counted using the Cell Counter tool by ImageJ. (**D**) Cells were seeded on 6-well plates and left to form monolayers. After reaching the desired confluency, wounds were scratched, and the Bortezomib-free media were replaced with medium containing 10% FBS and Bortezomib. The naïve cells were assessed using 20 nM of Bortezomib, and the DU-145 RB60 cells were assessed under the influence of 60 nM Bortezomib. After replacement of the media, photographs were taken, and the wound closure rate was determined by capturing images at the specific time-points of 0, 24, 48, and 72 h. The photographs were analyzed using the ImageJ Manual Wound Healing Size tool and the subsequent analysis was performed using Microsoft Office Excel and GraphPad Prism 8.

with or without the presence of Bortezomib, and we found that only the resistant clone achieved healing of the wound without changes as compared to the control (Fig 1D). The untreated naïve clone (control) managed to cover 60% of the wound in the first 24 h and almost 95% at the 48-hour time point. After treatment with 20 nM of Bortezomib, the naïve clone failed to fully heal the scratches (even after 72 h) and the wound closure percentages fell to 5% in the first 24 h, followed by an ~25% and an ~75% percentage at the 48 h and 72 h time points, respectively. Treated and untreated DU-145 RB60 cells exhibited almost identical healing patterns; following an initial delay at the 24-hour key point (compared to the control

**Table 3. $IC_{50}$ Calculation of naïve DU-145, DU-145 RB60, and DU-145 RB60U cells.** The data from crystal violet assays were analyzed using GraphPad Prism 8, and the calculated $IC_{50}$ values are presented here. The resistant cells exhibited a 5-fold increase in Bortezomib tolerance after 24 weeks of Bortezomib presence, which was augmented more following another 24 weeks of drug presence, reaching a more than 10-fold increase compared to the naïve clone. Additionally, to some extent, cross-resistance to Carfilzomib was observed; the DU-145 RB60 clone achieved 4-fold Carfilzomib resistance compared to the naïve clone after 24 weeks and an almost 6-fold change after another 24 weeks. The long-deprived clone maintained its acquired resistance to both inhibitors during the 24-week monitoring. Regarding resistance to doxorubicin, all three cell clones tested exhibited a similar response to doxorubicin, regardless of resistance to Bortezomib. The experiments were repeated in triplicate, and for the $IC_{50}$ calculation, the built-in model from GraphPad Prism 8 was used.

| | $IC_{50}$ (nM) | | | | $IC_{50}$ ratio (RB60:naïve) | |
|---|---|---|---|---|---|---|
| Cell clone | Naïve | RB60 (1 w) | RB60 (24 w) | RB60U (24 w) | Week 1 | Week 24 |
| **Bortezomib** | 12.19±1.06 | 65.87±1.96 | 125.5±1.02 | 63.05±2.10 | 5.40 | 10.29 |
| **Carfilzomib** | 8.84±1.17 | 35.74±1.56 | 52.00±1.51 | 33.06±1.08 | 4.04 | 5.88 |
| **Doxorubicin** | 341±2.17 | 342±1.43 | 340±2.07 | 342±1.67 | | |

group), they had almost fully healed the scratch in the first 48 h (both ~95%). Thus, we deduced that resistant DU-145 RB60 cancer cells appeared to fully restore and maintain all the main biological characteristics in the presence of Bortezomib without significant changes compared to the naïve, untreated DU-145 clones.

## Bortezomib-resistant clones successfully restore the Ubiquitin-Proteasome system activity, even on high drug doses

In cancer cells, Bortezomib successfully inhibits protein degradation and promotes the accumulation of polyubiquitinated proteins. To investigate the response of the resistant clone to Bortezomib, we assessed the polyubiquitination of the total proteins in both resistant and naïve DU-145 clones by western blot analysis (Fig 2A–2C). Time-course experiments showed that the naïve DU-145 clone accumulated polyubiquitinated proteins after incubation with 20 nM Bortezomib for 24 h, while on the DU-145 RB60 clone, the same effect was not observed (Fig 2B). The DU-145 RB60 cells accumulated ubiquitinated proteins only after incubation with >180 nM Bortezomib (Fig 2C).

This suggests that the resistant cells, despite their tolerance to the drug, exhibit an accumulation of ubiquitinated proteins at higher concentrations. Subsequently, we chose the 24-hour time point and the dose of 60 nM Bortezomib to conduct the signaling investigation experiments. At this specific time point, the naïve cells were affected by the drug, whereas the resistant clone remained intact and the RB60U cells showed mild accumulation of polyubiquitinated proteins.

In many similar studies, the molecular mechanism of Bortezomib resistance relies on a dramatic overexpression of the PSMB5 protein [17]. To clarify whether this mechanism might also be active in our resistant cancer cell line, we determined the expression of proteasome subunit β5 on DU-145 RB60 cells and the naïve cells by western blot analysis (Fig 2D). Overall, higher expression of this proteasome subunit was seen in the DU-145 RB60 cells compared to the naïve DU-145 cells. Protein levels of the β5 subunit were higher (P < 0.0001) in the DU-145 RB60 clone (almost double) with or without the presence of Bortezomib. Incubation with Bortezomib for 24 h slightly decreased the total amount of detectable PSMB5; however, this was not a statistically significant difference but rather an observation (Fig 2E). This decrease could be explained by an increased PSMB5 degradation rate following its blockage by Bortezomib. To clarify if the increased β5 subunit expression observed in the resistant clones would lead to a subsequent induction of elevated proteasome activity, we determined the basal chymotrypsin-like (ChT-L) activity on both resistant and naïve clones. Our results showed that the ChT-L activity of the DU-145 RB60 clone (even after Bortezomib short-term deprivation) was significantly higher than that of the naïve DU-145 clone (an almost 3-fold change),

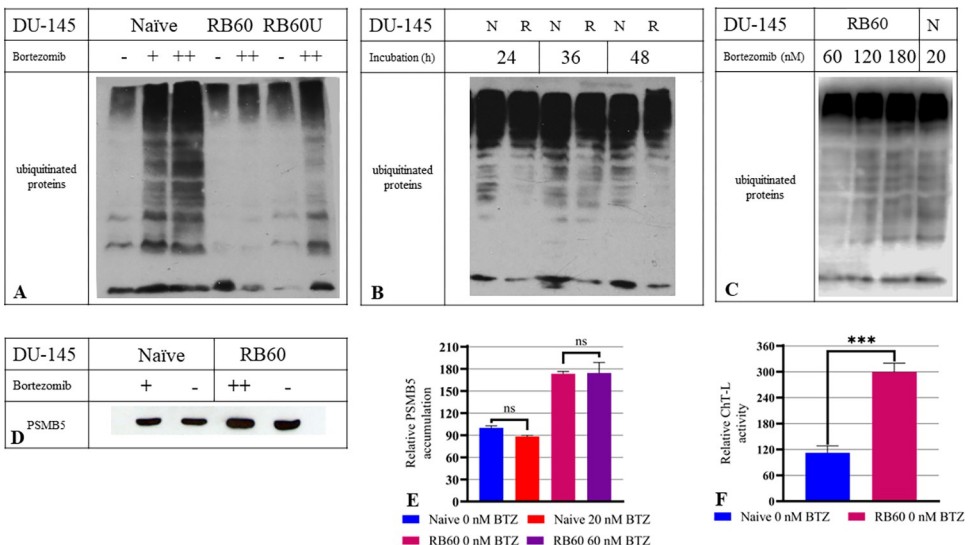

**Fig 2. Ubiquitin-Proteasome system assessment in naïve DU-145, DU-145 RB60U, and DU-145 RB60 resistant cells.** We examined the function of the UPS with Western blots of polyubiquitinated proteins. (**A**) An initial dose-response experiment was conducted on all three clones (naïve DU-145, DU-145 RB60U, and DU-145 RB60) following 24 h of incubation with Bortezomib. (**B**) Time-course experiments verified the stable ubiquitination levels at the key intervals of 24, 36, and 48 h following incubation with 20 nM Bortezomib, validating the 24-hour time point as an adequate time point to study effects on main signaling pathways. (**C**) Further dose-response experiments verified the susceptibility of DU-145 RB60 cells to Bortezomib at concentrations greater than 180 nM. (**D**) PSMB5 is the most ubiquitous proteasome subunit exhibiting chymotrypsin-like (ChT-L) activity. Following treatment with Bortezomib, we used Western blot analysis to estimate the PSMB5 protein levels. (**E**) The results were quantified using ImageJ, and to compare the groups, two-tailed t-tests were performed in GraphPad Prism 8. Between naïve and resistant cells, statistically significant differences were documented, regarding PSMB5 accumulation (P<0,0001), while the slight difference that occurred after treatment with Bortezomib were not found to be significant. (**F**) We estimated the ChT-L activity of naïve DU-145 and DU-145 RB60 cells using fluorometry. The resistant cells exhibited almost 3-fold increased proteasome activity compared to the naïve clone (P<0,0001). The results were quantified using ImageJ, and to compare the groups, two-tailed t-tests were performed in GraphPad Prism 8 All experiments (**A-E**) were performed in triplicate, and wherever bar charts are shown; the bars represent the means, and the error bars are the SEM.

suggesting the emergence of an acquired Bortezomib resistance mechanism that is independent of the constant drug presence (Fig 2F).

## Resistant cells evade apoptosis induced even by high Bortezomib doses

To investigate the mechanism by which the resistant clone manages to avoid drug toxicity and survive at higher Bortezomib concentrations, we determined the apoptotic rate in both clones by flow cytometry using the Annexin V/Propidium Iodide assay. Incubation of naïve DU-145 cells with 60 nM Bortezomib led to the induction of apoptosis at a significant degree (P < 0.0001) (Fig 3B) compared to the untreated group (Fig 3A), an effect that was not observed on the DU-145 RB60 clone (Fig 3D). The comparison was performed with a two-tailed t-test, comparing the Annexin-V populations among the different samples. The DU-145 RB60 clone appeared to be unaffected by low and medium doses of Bortezomib and exhibited the same apoptotic rate as untreated naïve DU-145 cells (Fig 3C and 3D). Both early and late apoptotic rates of DU-145 RB60 cells remained at least 6-fold lower than those of naïve cells, suggesting that these cells were capable of suppressing cell death induction mechanisms and eventually evading apoptosis.

## Resistant cells overthrow cell cycle arrest induced by Bortezomib

The effects of Bortezomib on the main apoptotic signaling molecules have been long-established [4–7]. However, the consequences of the drug on cell cycle regulation and the

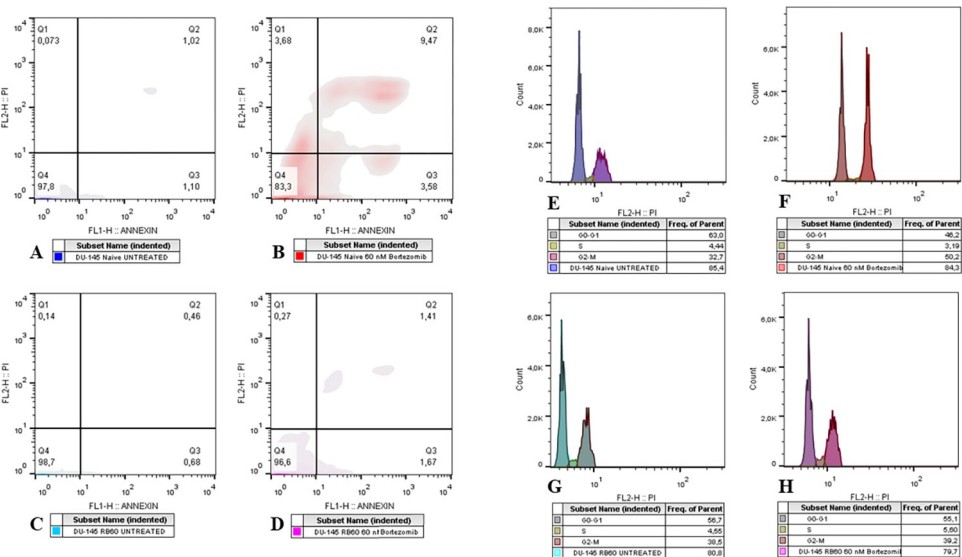

**Fig 3. Apoptosis and cell cycle assays using Annexin V-FITC conjugated and propidium iodide.** (**A-D**) Cells were treated with Bortezomib for 48 h and subsequently trypsinized and stained with Annexin V/PI. Flow cytometry analysis using Annexin V-FITC and PI indicated that the DU-145 RB60 cells I have fully restored their apoptotic rate to basal levels (**A**). Treatment with 60 nM of Bortezomib greatly induced apoptosis, diminishing viability by 15% in the first 48 h in naïve DU-145 cells (**B**), while the same treatment only slightly increased the percentage of apoptotic DU-145 RB60 cells (**D**). (**E-H**) Cells were treated with Bortezomib for 24 h following 48 h of drug deprivation from the DU-145 RB60 cells. After trypsinization, the cells were fixed using ice-cold methanol and later stained with propidium iodide. (**F**) G2 arrest was observed on the treated naïve clone, while the RB60 clone was found to have fully abolished Bortezomib effects (**G**). The RB60 cells did not indicate any major changes after Bortezomib treatment (**H**) and had similar phase distributions with the untreated naïve clone (**E**).

interrelationship between those two mechanisms are not fully understood. In general, Bortezomib inhibits cell cycle progression mainly by dysregulating the turnover of various cell cycle regulators, such as cyclins and CDK-associated molecules [46, 47]. Analysis of the cell cycle using PI indicated that at the basal state of naïve DU-145 cells (untreated) and DU-145 RB60 cells (treated), the cell cycle progresses undisrupted (Fig 3E, 3G, and 3H, respectively). Treatment with Bortezomib causes cell cycle arrest, and most of the cells exited the $G_1$ phase, entering $G_0$, and those already in the $G_2$ phase never progressed to mitosis (Fig 3F), a finding already noted by others [25, 37, 47–49]. Cell cycle inhibition has already been associated with p21 and p27 accumulation following Bortezomib treatment in vivo, and the restoration of this system in the resistant clones studied surely allows survival and proliferation.

The accumulation of p21$^{\text{waf1/kip1}}$, a known cell cycle inhibitor, has been shown to correlate with Bortezomib's action [6, 20, 50, 51]. Incubating the cells with Bortezomib led to an increase in the nuclear levels of p21 protein in a dose-response manner (Fig 4A) and in a time-course manner, which showed a peak after 24–36 h of incubation. It is well known that the nuclear localization of p21 mainly causes cell cycle arrest, while its cytoplasmic presence has been associated with antiapoptotic activity [52].

A concentration of 20 nM was adequate to induce the accumulation on naïve DU-145 cells, while the same was not observed on RB60 cells. The turnover rate of p21 is already known to be regulated through proteasomal degradation, a mechanism disrupted in DU-145 cells, leading to the sub sequence of p21 protein accumulation. The resistant cells were able to downregulate p21 accumulation levels independently of the drug presence, and this characteristic remained after long-term deprivation of the drug, as shown on the RB60U cell clone (Fig 4A). The p21 protein was detected inside the nucleus of naïve DU-145 upon treatment with Bortezomib, verifying its proapoptotic role (Fig 4B).

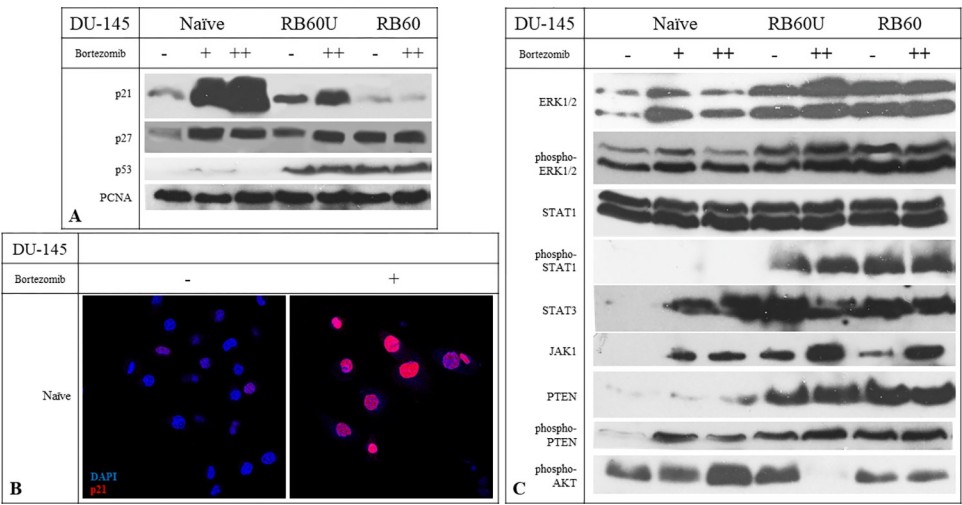

**Fig 4. Effects of Bortezomib on main cell cycle regulators and signaling pathways.** (A) Western blot analyses of key cell cycle proteins p21, p27, and p53 and the proliferation marker PCNA. The experiments were conducted after 24 h of Bortezomib incubation, following a 48 h Bortezomib deprivation of RB60 cells. The + corresponds to the low dose of 20 nM Bortezomib, and the ++ corresponds to the medium dose of 60 nM. The cells were lysed using ice-cold lysis buffer, and protein concentrations were determined using the Bradford assay. The same amounts of total protein were loaded on 12% SDS-PAGE gels, and the transfer was performed using the semidry system. (B) Immunocytochemical staining of naïve DU-145 cells with antibodies against p21 and DAPI to visualize DNA content to assess the localization of p21. Following treatment with 20 nM Bortezomib, the nuclear localization of p21 was verified, indicating the pro-apoptotic role of p21 rather than its pro-survival cytosolic presence. (C) The main pro-survival and proliferation pathways were assessed following the incubation of cells with Bortezomib. All experiments were conducted after 24 h of Bortezomib incubation following a 48-hour drug deprivation from the resistant clones, and the protein quantity was confirmed using the Bradford assay.

The cell cycle regulator p27$^{Kip1}$ was not found to follow the same accumulation pattern as p21 in the resistant clone studied. After incubation with Bortezomib, the p27 protein maintained its high nuclear levels, both in naïve and resistant cells. Deprivation of Bortezomib for 24 h reduced the intracellular levels of p27 on DU-145 resistant cells imitating an untreated naïve cell (Fig 4A). Since the p27 protein is degraded through the UPS, the similar protein levels detected might indicate that the resistant clone that emerged in our study has somehow managed to balance the pro-apoptotic signals descending from the accumulation of cell cycle inhibitors with survival-promoting molecules and antiapoptotic proteins.

The p53 protein was also assessed to investigate potential alterations in its expression and activity. Levels of p53 protein were found to be elevated in the resistant clones studied, a finding observed in both RB60 and RB60U cells (Fig 4A). Incubation of naïve cells with Bortezomib led to a slight increase in its presence, although the accumulation inside the resistant clones was incomparable.

Finally, the presence of PCNA was assessed using a western blot to detect whether the substitution of the cell's ability to multiply at a normal (compared to naïve cells) rate relies on the overexpression of replication-associated molecules that may surpass the inhibitory signaling cascades. However, PCNA showed similar expression patterns in both the experimental clones, with an expected decrease in its accumulation on DU-145 cells after 24 h of treatment with Bortezomib (Fig 4A).

## JAK/STAT, Akt, and MAPK mediate the survival of the resistant clone

Following the results indicating suppression of cell cycle inhibitors and the restoration of proliferative activity in resistant clones, we examined key signaling molecules controlling these functions upstream in the signaling cascades.

The most interesting findings concerning cell survival were those associated with MAPKs. The ERK1/2 protein kinases were found to be overexpressed on the resistant and Bortezomib-deprived clones, compared to the naïve clone (Fig 4C), and additionally, the phosphorylation patterns were greatly different. After incubation with Bortezomib, the naïve cells were found to phosphorylate ERK1/2 during the first 24 h at low Bortezomib doses (<20 nM); however, after prolonged incubation or exposure to moderate or high doses of the drug (20–60 nM), phosphorylation was abrogated at 36–48 h. The resistant clones exhibited a different activation profile. Both MAPKs were found to be phosphorylated during Bortezomib incubation, and their activation only declined after a remarkably high dose of Bortezomib was added, reaching the milestone of 240 nM. Between untreated and treated DU-145 RB60 cells, no significant differences in phosphorylation were observed, indicating a permanent ERK1/2 activation independent of Bortezomib that regulated the new equilibrium (Fig 4C).

The JAK/STAT signaling pathway was also studied. JAK1 protein was found to be overexpressed on the DU-145 RB60 and RB60U clones upon Bortezomib incubation (Fig 4C). JAK1 is known to phosphorylate members of the signal transducer and activator of the transcription protein family (or STAT), which facilitates the transcription of surviving-associated genes. While the total STAT1 protein was detected at similar levels among naïve and resistant cells (Fig 4C), the DU-145 RB60U and DU-145 RB60 clones induced higher levels of phosphorylation (Fig 4C), a response pattern already connected to evasion of Bortezomib-induced apoptosis [53]. The total STAT3 was also analyzed and found to be present in all cell types; however, the basal expression levels on the DU-145 naïve cells were significantly lower than those of the treated and resistant cells (Fig 4C). The JAK/STAT pathway is a major survival and differentiation pathway, active in many aggressive forms of cancer, and therefore, its activation in the resistant cells is a reasonable finding.

The activity of PTEN, a phosphatase inhibiting the AKT/mTOR pathway, was also studied, as was the activation of AKT at the key time point of 24 h. The activated forms of PTEN and AKT had opposite actions: the phosphorylated PTEN promoted apoptosis, while the phosphorylated AKT favored survival. DU-145 RB60 and RB60U cells were found to express higher levels of PTEN compared to the naïve cells (Fig 4C) however, a study of its phosphorylation indicated that on the resistant clones, the dominant form is the dephosphorylated and hereby inactive (Fig 4C). The phosphorylated form of PTEN was significantly higher in the DU-145 naïve cells following Bortezomib treatment, exhibiting a dose-response pattern, while the total PTEN levels, both phosphorylated and dephosphorylated, were lower at the baseline conditions.

Even though the activated PTEN is likely to induce AKT dephosphorylation, AKT was found activated following treatment with Bortezomib on the naïve cells, showing a dose-response pattern (Fig 4C). A lack of AKT activation was observed on DU-145 RB60U cells after 24 h of Bortezomib treatment, which was quite controversial, whereas the levels of phosphorylated AKT did not change on the RB60 clone. High levels of AKT have also been mentioned by other researchers [54]. At the same time, the ERK kinase is vastly phosphorylated on both RB60 and RB60U clones, indicating an alternative way to affect survival and suppress proapoptotic signals.

## Bortezomib-resistant cells utilize autophagy as a substitute for the impaired proteasome-ubiquitin system

Preliminary data from other researchers have indicated an induction of autophagy in Bortezomib-resistant cells as a balancing mechanism to obtain nutrients from damaged molecules, banish accumulated non-functional proteins, and suppress the activation of pro-apoptotic

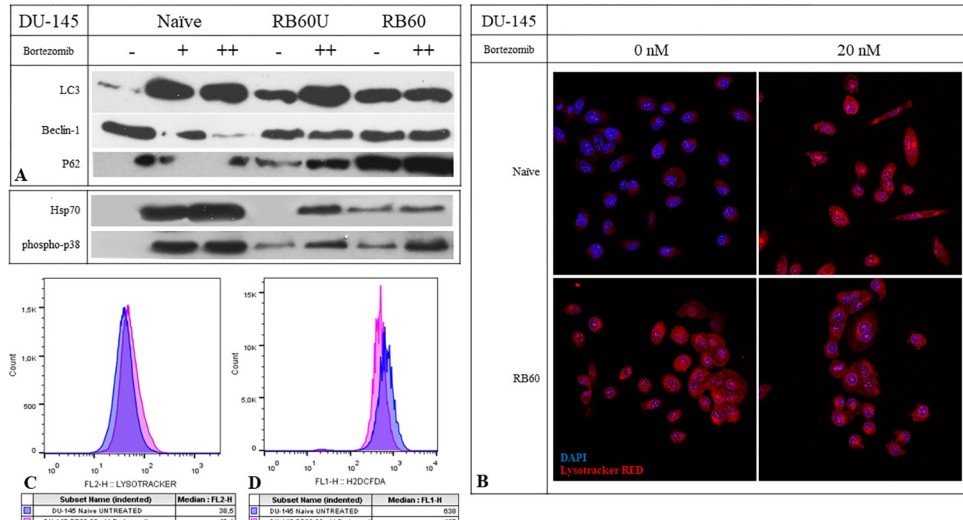

**Fig 5. Autophagy and oxidative stress assays using western blots, flow cytometry and confocal microscopy. (A)**
Western blot analyses of key proteins regulating autophagy (LC3, Beclin-1, p62) and stress markers (Hsp70 and
phosphor-p38). The experiments were conducted after 24 h of Bortezomib incubation following a 48 h Bortezomib
deprivation of RB60 cells, as previously noted. The + corresponds to the low dose of 20 nM Bortezomib, and the +
+ corresponds to the medium dose of 60 nM. **(B)** Staining of naïve DU-145 cells with Lysotracker RED, which stains
acidic proteins, following treatment with 20 nM Bortezomib. The cells were cultured on coverslips and stained
(without fixation) with Lysotracker RED for 15 min at 37˚C, followed by confocal imaging. **(C)** Flow cytometry
analysis of Lysotracker RED inside naïve and resistant live cells. The cells were trypsinized and subsequently stained
with Lysotracker RED for 45 min followed by analysis using a FACS Calibur flow cytometer. **(D)** Flow cytometry
analysis of ROS generation using $H_2DCFDA$. The cells were incubated for 24 h with Bortezomib and then trypsinized.
During the staining procedure, they were maintained inside the culture medium to avoid heat shock and starvation
stress. Staining was performed at 37˚C and the samples were rinsed with PBS and analyzed using a FACS Calibur flow
cytometer.

pathways [25, 27, 31]. Additionally, the joint assembly of proteasomes and autophagosomes
and the emerging process called proteaphagy have been documented to increase after the
administration of proteasome inhibitors, potentially in an effort to eliminate blocked proteasomes [55]. To assess these phenomena, the lysosomal activity was studied using the alkaline
dye Lysotracker RED, which can visualize (Fig 5B) and quantify (Fig 5C) acidic protein content. Lysotracker RED indicated increased activity inside the resistant cells, which was also
quantified using flow cytometry. Additionally, by using a western blot analysis, the main
autophagy biomarkers LC3 a/b, Beclin-1, and p62 were examined (Fig 5A1).

The LC3 protein is a major autophagy marker. Compared to the untreated cells, the administration of Bortezomib generally increased LC3 accumulation on both naïve and resistant
clones. However, the three clones, DU-145 RB60, DU-145 RB60U, and DU-145 RB60, differed
vastly at the basal levels of LC3 expression. Basal LC3 expression was the lowest in the naïve
cells, and the long-term deprivation of Bortezomib on DU-145 RB60U cells led to increased
LC3 levels, which were slightly lower than those of the resistant RB60 clone. Upon treatment
with the drug, the naïve cells exhibited induction of LC3, while in the resistant clone, LC3
expression remained relatively stable. The RB60U clone showed an intermediate pattern
between the two states, exhibiting basal LC3 levels similar to those of the resistant clone, indicating the permanent alteration of cell functions, while at the same time exhibiting an LC3
overexpression as in the naïve clone. Autophagy seems to be regulated by the same pathways
that allow the emergence of resistance since the basal autophagy levels of the RB60U and RB60
clones were almost identical (Fig 5A1).

Beclin-1 is a key molecule in the crosstalk between autophagy and apoptosis, credited with an antiapoptotic role [56, 57]. High expression of Beclin-1 leads to autophagy activation, and its upregulation is believed to be connected to Bortezomib resistance. Knockdown of Beclin-1 has been shown to reverse Bortezomib resistance in cancer cells by inducing apoptotic cell death [34]. The DU-145 naïve cells were found to have increased basal levels of Beclin-1 that declined following treatment with Bortezomib, justified by the simultaneous apoptosis induction. The DU-145 RB60 and RB60U cells maintained relatively similar levels of Beclin-1 during treatment with the drug, which were higher than those of the treated naïve cells. The stable accumulation of Beclin-1 on the resistant clones upon exposure to Bortezomib explained the substitution of UPS impairment by proteophagy and the maintenance of a low apoptotic rate, similar to that of the naïve cells. A dose of 60 nM Bortezomib was found adequate to significantly downregulate Beclin-1 accumulation on naïve DU-145 cells, while the same dose on the long-deprived clone DU-145 RB60U did not provoke any changes in its presence (Fig 5A1).

The p62 protein acts as a cargo receptor, transporting ubiquitinated proteins to the autophagosomes and assembling structures, known as proteaphagosomes [55]. During the process of autophagy, its levels decrease through subsequent degradation. The naïve cells were observed to maintain low p62 levels even after Bortezomib treatment, showing only moderate changes. However, the resistant clones showed a different pattern. The p62 protein was overexpressed inside the resistant DU-145 RB60 and DU-145 RB60U clones, indicating long-term changes in the way cells utilize their nutrients and maintain their homeostasis (Fig 5A1). The accumulation of p62 may be the key mechanism of autophagy upregulation as an alternative pathway to substitute proteasome-mediated protein degradation.

## Resistant cells have lower oxidative stress levels and are less prone to drug-induced damage

Oxidative stress is a proposed mechanism of action for many chemotherapies, acting by damaging vital biomolecules and inducing cell death. Bortezomib is believed to exert a similar action, because of the disruption of cell energetics, the dysregulation of degradation pathways, and the subsequential accumulation of free radicals resulting from the failure of the counteractive homeostatic mechanisms [9, 58–61]. The oxidative stress levels of naïve and resistant cells were assessed using the $H_2DCFDA$ assay. The resistant cells indicated lower intracellular Reactive Oxygen Species (ROS) levels compared to the naïve clone (Fig 5D) Specifically, impairment of the UPS system is believed to increase oxidative stress on cancer cells, and this effect was verified in our study; however, the resistant cells are found to achieve lower ROS levels than those achieved by the naïve untreated cells, possibly by inducing antioxidative defense mechanisms to survive under regular doses of Bortezomib.

We further examined the effects of Bortezomib on proteins related to stress conditions, such as heat shock proteins [62] and p38 MAPKs [63]. The heat shock protein 70 (Hsp70) was analyzed using western blots and indicated a dose-response induction in the presence of Bortezomib with a peak at 24 h. The DU-145 RB60 cells were capable of suppressing its accumulation, while the DU-145 RB60U cell clone exhibited a naïve-like phenotype regarding Hsp70 (Fig 5A2). The p38 MAPK belongs to a distinctive class of MAPKs that are activated through phosphorylation during stress conditions induced by radiation and drugs [64]. Bortezomib induces such an effect on naïve cells, yet the resistant clone was once again immune to it, maintaining low phosphorylation levels (Fig 5A2). The two proteins mentioned, p38 and Hsp70, are thought to be correlated, with the molecular chaperon Hsp70 acting as a molecular chaperon mediating the nuclear translocation of the activated p38 [65, 66].

## Discussion

The proteasome is a key functional structure in cancer cell homeostasis, and the abrogation of its function by proteasome inhibitors (PIs) is a long-established therapeutic approach already applied to several types of malignancies. However, the emergence of resistance against PIs is a major therapeutic drawback. Our study elucidates many aspects of the acquired resistance of prostate cancer cells, pointing out potential ways to target them and reverse them. Besides the research for treatment of prostate cancer, for whom DU-145 cells are a well-established model, Bortezomib is used to treat hematological malignancies such as mantle cell lymphoma [67] and multiple myeloma [68], while at the same time, it has shown little efficacy against clonal cells of the Myelodysplastic Syndromes (MDS) [69]. Even though clinical trials of Bortezomib have been conducted [13, 14, 38], the weak response of prostate cancer to Bortezomib and the rapid development of resistance have limited further research.

The model of this study managed to acquire resistance against the PI Bortezomib, and the resulting cell phenotype was suspected to have a strong genetic basis since it remained stable for a prolonged period of time. The bottleneck effects caused by Bortezomib-induced massive cell deaths, combined with the dysregulation of other main signaling pathways on the surviving cells and the accumulation of somatic mutations, led to the generation of a Bortezomib-resistant cell line with increased cross-resistance to second-generation PIs such as Carfilzomib. Additionally, the resistance phenotype was documented to apply only to proteasome inhibitor resistance, since both cell lines (resistant and non-resistant) were susceptible to doxorubicin and did not exhibit alterations at their $IC_{50}$. The apoptotic rate of the newly emerged resistant cell clone gradually diminished, finally reaching that of naïve cells, and the same was observed regarding the resistance to cell cycle regulators. Three major cell cycle regulators were assessed: p21, p27, and p53, all of which had already been studied previously and connected to the effects of Bortezomib treatment [4, 20, 70]. DU-145 cells following incubation with PIs were found to dramatically increase p21 protein levels, which we observed to be downregulated on the resistant clones, while an increase of p27 and p53 was also observed, as has been previously mentioned. Although the levels of p27 remained relatively high on our resistant clones, compared to the basal state of naïve cells, in our opinion, the most interesting finding of this study is related to the p53 protein, which is an important part of the p21/p53 axis [70]. The p53/survivin system has already been associated with Bortezomib resistance through the abrogation of Bortezomib-induced apoptosis in many types of cancer. Wild-type p53 makes cells more susceptible to Bortezomib, while mutant variants of p53 have a strong association with Bortezomib resistance [71]. p53 overexpression on our established resistant clone may have resulted from a mutation or the cell's normal response to Bortezomib-induced stress and UPS impairment. The latter, mainly observed in resistant clones, may be warded off by a different mechanism, such as the assessed autophagy. Besides the high p27 and p53 expression levels in resistant cells, their ability to enter the cell cycle was restored. The combination of a wild-type p53 and functional p21 renders the second a genome guardian, while in the absence of or presence of a mutated p53, p21 can induce genomic instability [70]. Our findings support this idea, pointing out that the restoration of p21 on DU-145 Bortezomib-resistant cells could reverse resistance.

The resistant cells' ability to thrive in the presence of Bortezomib was further analyzed by looking at the key signaling molecules controlling cell survival and proliferation. Our study indicated that Bortezomib resistance in our clone heavily relies on the constant activation of MAPKs, the JAK/STAT pathway, and the activation of AKT. These molecules are already known to be active in many aggressive forms of human cancer and to counteract the proapoptotic signals induced by Bortezomib, and their exogenous inhibition could reverse the resistant

phenotype [25]. Indeed, the phosphorylated forms of STATs have been found to counteract the proapoptotic effects of Bortezomib in ovarian cancer, leading to the emergence of Bortezomib resistance [53]. STAT3 has also been found to control the expression of β subunits of the 26S proteasome [72]; therefore, the high STAT3 levels found in DU-145 RB60 cells could regulate the expression of *PSMB5* to counteract the accumulation of dysfunctional proteasomal subunits induced by Bortezomib binding. Phosphorylation of STATs as well as of other survival-promoting transcriptional factors can be mediated through ERK1/2, which in our resistant clones were found to be constantly activated. Inhibition of ERK1/2 phosphorylation by MEK inhibitors has been reported to reverse Bortezomib resistance in the SKM-1 MDS cell line [25]. In parallel, activation of the AKT pathway indicated an induction in the resistant clones. Moreover, in our study, the expression of the suppressing protein PTEN was found to be higher on both DU-145 RB60 and RB60U clones; however, the dominant form of PTEN was the dephosphorylated one. High PTEN levels have already been documented as a consequence of Bortezomib treatment, eventually leading to apoptosis [73]. The phosphorylated form of PTEN can stabilize the p53 protein, and PTEN transcription can be induced by p53 [74] therefore, the crosstalk between p53 and PTEN is of major significance [75].

Additionally, the regulation of the other major protein degradation pathway, namely autophagy, was assessed and found to be upregulated in the resistant cells. It has previously been shown that autophagy may be a key substitute for peptide degradation when the UPS system fails [33, 34]. The degradation of inhibited proteasomes at the autophagosomes is probably a key point in eliminating the blocked subunits [27] and this might be more easily induced when inhibition is irreversible with the newer PIs. In addition, the overall induction of autophagy may fulfill the need for amino acid recycling, as was shown by the constant p62 accumulation. The equilibrium between apoptosis and autophagy, which can also lead to cell death, is mediated through proteins that were found to be upregulated, mainly Beclin-1 and MAPK p38, which were found phosphorylated following Bortezomib treatment. The p38 protein has been credited with multiple roles, among them the regulation of autophagy [76]. The increased basal levels of p38 in its phosphorylated form, combined with the induction of autophagy, could imply that in our clone, p38 could function as a mediator of resistance. Thus, inhibiting p38 could, even partially, reset the cell's ability to withstand the presence of Bortezomib. Hereby, we show for the first time that autophagy is upregulated in a Bortezomib-resistant prostate cancer cell line, a finding confirming a similar phenotype to Bortezomib-resistant Multiple Myeloma cells as shown by Lernia et al. in 2020 [77].

A crucial element closely related to the many proteostasis subsystems is Hsp70, a protein that functions as a molecular chaperone. Hsp70 is believed to have a dual role in assisting both protein refolding, and leading damaged, misfolded, or non-functional proteins to the 26S proteasome subunit after ubiquitin ligation [78]. A permanent increase in Hsp70 accumulation could also function as a mechanism of resistance; however, such an effect was not observed in our study. Hsp70 levels exhibited great augmentation after Bortezomib treatment of naïve cells in a dose-response manner, despite the low basal levels observed in RB60 and even RB60U cells. Hsp70 levels of long-deprived DU-145 RB60U cells were identical to those of naïve untreated cells, while the presence of Bortezomib did not significantly alter Hsp70 expression, compared to the stable (and low) expression observed in the DU-145 RB60 clone. Induction of Hsp70 has been documented to suppress autophagy [79, 80], therefore, despite its protein refolding role, given that our clone mainly achieves resistance through autophagy regulation, Hsp70 protein remains at low levels.

Besides its role in proteostasis, the Hsp70 protein has also been associated with stress conditions inside the cells. The downregulation of Hsp70 observed in the resistant cells, combined with the fact that these cells thrive in a Bortezomib-rich environment, led us to study oxidative

stress levels. Proteasome inhibitors induced the elevation of ROS levels as a result of damaged protein accumulation, an increased need for energy (to biosynthesize new proteins and replenish the damaged ones), and dysregulation of cellular homeostasis [42, 61]. Surprisingly, ROS levels in the resistant cells were found to be lower, even than those of the untreated naïve cells, suggesting that the resistant cells had rather fully reversed the oxidative stress damage induced by Bortezomib. These observations, combined with the finding that the resistant clones cultured in our laboratory exhibited increased ChT-L activity, lead us to the conclusion that increased proteasome proteolytic activity can lower ROS levels, which may function as a survival mechanism for the drug-resistant cells. Stress-induced cell death is a long-suspected mechanism of action of the PIs, also explaining the increased rate of apoptosis-induction observed in cancer cells compared to normal cells following exposure to Bortezomib [59]. This observation of oxidative stress can be used as a therapeutic agent, acting synergistically with classical anticancer drugs. Oxidative stress has been confirmed to act as an intracellular signal that promotes the transcription of survival-related genes through the synthesis and activation of STAT3 [81–83] and given the established role of STAT3 on PSMB5 expression [72], exhaustion of antioxidant defense mechanisms could dysregulate the equilibrium, limiting Bortezomib efficacy. The administration of classic chemotherapeutic agents, such as anthracyclines and cisplatin, in patients that eventually developed resistance to PIs could be resumed if inducers of cellular oxidative stress could be concurrently used as therapeutic agents. The potential depletion of the redox homeostasis system capacity would render the cells more susceptible to proteostasis impairment [84], and thus, the emergence of resistance would become less likely. Furthermore, the mechanisms underlying Bortezomib resistance (autophagy, oxidative stress, and MAPK signaling) are already associated with an important role in metabolism [85–87] therefore, the metabolic alterations observed in Bortezomib-resistant cells shall be identified and extensively studied. Autophagy and oxidative stress could act both as markers and targets of a dysregulated cancer cell that slowly drifts from a UPS controlled protein degradation process to an autophagy mediated process. The pathways controlling autophagy as well as the amino acid synthesis and degradation pathways must be assessed given the role Bortezomib has on this specific homeostasis subsystem and possibly designating Bortezomib resistance as a phenomenon with a strong metabolic basis.

## Supporting information

**S1 Fig. Apoptosis and cell cycle analysis of DU-145 RB60U Cells.** (**A, B**) The DU-145 RB60U cell clone maintains the apoptosis evasion observed on the DU-145 RB60 cells after a 24-week drug deprivation with a slight increase of its apoptotic rate. (**C, D**) Following the same procedures as in naïve and DU-145 RB60 cells, the cell cycle of DU-145 RB60U cells was analyzed using PI and indicated a mild G2 arrest following incubation with 60 nM of Bortezomib for the first time after 24 weeks. The subsequent cell cycle inhibition does not result in apoptosis as has been shown by the rest experiments assessing apoptosis compared to the naïve DU-145 that within 48–72 h undergo apoptosis.
(TIF)

**S2 Fig. Chemotactic assay of DU-145 naive and DU-145 RB60 cells using Boyden chambers.** Cells were transferred into a chamber containing serum-free medium with or without Bortezomib. The chambers were placed inside microplates' wells containing medium supplemented with 20% FBS and left to migrate for 24 h. The DU-145 cells, when exposed to Bortezomib (20 nM), decreased their migration rate. Inhibition of migration was also observed when Bortezomib was added to the lower compartment, indicating a chemorepellent role. The DU-145 RB60 cells were also repelled by Bortezomib (60 nM), while the presence of the drug

in the upper compartment induced migration towards the other side of the membrane, where Bortezomib was absent.
(TIF)

**S3 Fig. Wound healing assay of DU-145 naïve and DU-145 RB60 cells.** Cells were seeded on 6-well plates and left to form monolayers. After reaching the desired confluency, wounds were scratched, and the Bortezomib-free media were replaced with medium containing 10% FBS and Bortezomib. The naïve cells were assessed using 20 nM of Bortezomib, and the DU-145 RB60 cells were assessed under the influence of 60 nM Bortezomib. Compared to the control group, the DU-145 naïve cells' ability to heal wounds was heavily impaired by Bortezomib, while the same effect was not observed on the DU-145 RB60 cells. The resistant cells were able to completely heal the scratches after 72 h of incubation, and the same was achieved by the untreated naïve cells.
(TIF)

**S1 Raw images.**
(PDF)

## Author Contributions

**Conceptualization:** Kalliopi Zafeiropoulou, Georgios Kalampounias, Argiris Symeonidis.

**Data curation:** Georgios Kalampounias.

**Funding acquisition:** Argiris Symeonidis.

**Investigation:** Kalliopi Zafeiropoulou, Georgios Kalampounias, Spyridon Alexis, Daniil Anastasopoulos.

**Methodology:** Kalliopi Zafeiropoulou, Georgios Kalampounias, Spyridon Alexis, Argiris Symeonidis.

**Supervision:** Argiris Symeonidis, Panagiotis Katsoris.

**Validation:** Georgios Kalampounias.

**Writing – original draft:** Georgios Kalampounias, Panagiotis Katsoris.

**Writing – review & editing:** Argiris Symeonidis.

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
