## [Decision Letter · Decision Letter 0]

8 Nov 2023

PONE-D-23-23776Investigating the mechanisms underlying Bortezomib resistancePLOS ONE

Dear Dr. Katsoris,

Thank you for submitting your manuscript to PLOS ONE. After careful consideration, we feel that it has merit but does not fully meet PLOS ONE’s publication criteria as it currently stands. Therefore, we invite you to submit a revised version of the manuscript that addresses the points raised during the review process.

We thank you for your submission but the comments of Reviewer 1 need to be fully addressed for the paper to be accepted.

We look forward to receiving your revised manuscript.

Kind regards,

Laila Adel Ziko

Academic Editor

PLOS ONE

Journal Requirements:

Additional Editor Comments:

I received varying decisions from the reviewers so to be accepted, all the comments of reviewer 2 need to be addressed fully and appropriately in order for us to be able to accept the paper.

Reviewers' comments:

Reviewer's Responses to Questions

**Comments to the Author**

1. Is the manuscript technically sound, and do the data support the conclusions?

Reviewer #1: Partly

Reviewer #2: Yes

2. Has the statistical analysis been performed appropriately and rigorously? 

Reviewer #1: N/A

Reviewer #2: Yes

3. Have the authors made all data underlying the findings in their manuscript fully available?

Reviewer #1: Yes

Reviewer #2: Yes

4. Is the manuscript presented in an intelligible fashion and written in standard English?

Reviewer #1: No

Reviewer #2: Yes

5. Review Comments to the Author

Reviewer #1: - In this manuscript, Zafeiropoulou et al tried to investigate many aspects and phenotypes of the proteosome inhibitor (Bortezomib) resistant prostate cancer cell line. Despite presenting some preliminary interesting insights, there were many major concerns in the study:

1-Many of the outcomes of this study are a phenotype characterization of the prostate cancer bortezomib resistant cell line and not truly addressed the mechanistic pathways behind these resistance phenotypes as claimed by the aim of work and title of manuscript. The findings were a kind of association with Bortezomib resistance and not proved to be causative. The authors did not try to reverse the dysregulated markers observed in their results to see the corresponding effect on resistance and hence can build a causative relation.

2- The novelty of the study is limited as it is very well established that resistant cancer cells in general including previous studies on bortezomib resistance in hematological cancers have increased proliferation, decreased apoptosis , increased wound healing efficiency, and alteration of cell cycle regulation and other essential kinases investigated here by authors.

3- Point mutations or overexpression of β5 subunit of proteasome is a well-established cause of bortezomib resistance. This was also replicated by authors here with gene expression analysis showing overexp of β5 subunit in the resistant cell line compared to control cell line. However the authors did not add another novel mechanistic layer explaining the reason behind this increased expression( like genetic or epigenetic factors). It was not addressed by the authors whether this was the sole cause of resistance in their resistant cell line and then the other observed results are simply secondary consequences of this overexpression or there are additional novel mechanisms of resistance independent from β5 subunit overexpression. Knockdown of PSMB5 may be helpful to see if this can reverse the resistance completely or partially. In addition, sequencing of PSMB5 gene was also essential to investigate any point mutations which can cause hyperactive variant or hinder the binding with Bortezomib.

4- The authors did not discuss the rationale of using prostate cancer cells especially that bortezomib is used commonly in cancers of haematopoietic origin and will be more clinically relevant in hematological cell lines.

5- The authors did not test their resistant cancer cell line against anti-cancer drugs from another family (something other than proteosome inhibitors) to see whether these observed phenotypes are really specific to Bortezomib or it is more of a cancer cell resistance to treatment in general.

6-Title should be informative for the main message and outcomes (most significant result) of the study and not general description of the aim of study as authors do here.

7- The figures are so redundant and can be merged at many occasions in one figure with many panels and some others can be transferred as supplementary figures.

8- Western blots are lacking the essential loading controls ( figure 5A, 7, 9,10 and 11)

Reviewer #2: This study Investigated the mechanisms underlying Bortezomib resistance. The article is well-written.

I have no critical comments, but I hope to see soon further studies in in vivo models either in human or experimental animals. Also, I wished that the authors discussed other signaling pathways that might be involved in drug resistance and the possibility of Bortezomib to affect these pathways

6. PLOS authors have the option to publish the peer review history of their article (what does this mean?). If published, this will include your full peer review and any attached files.

Reviewer #1: No

Reviewer #2: **Yes: **Mohamed Z. Gad

---

## [Author Response · Author response to Decision Letter 0]

20 Nov 2023

Dear Dr. Ziko, 

Thank you for your email dated November 8, 2023, enclosing the reviewers’ comments. Our team has carefully reviewed all comments and revised our manuscript accordingly. 

Hereby, we will be answering one by one the constructive comments made during the peer review process, and as an attachment, we will send to you both a marked-up version of our revised manuscript, tracking all changes made, as well as the unmarked, official version. 

Additionally, we have uploaded a.pdf file containing the original Western Blot images with the appropriate annotation as a supporting information file on your platform. 

We hope the revised manuscript version is suitable for publication in your journal, and we look forward to hearing from you in due course.

Sincerely,

Dr. Panagiotis Katsoris

Professor

Department of Biology, University of Patras

Response to Reviewer 1:

Dear Sir/Madam,

Thank you for your review of our paper. Your comments really helped us to improve our manuscript and strengthen our findings and the way they are presented. We have answered each of the points below.

Regarding the General Comments to the Author:

We fully revised our manuscript, making our style more intelligible, correcting mistakes and changing key aspects of it such as our title, the arrangement of the figures and in many cases, the way we present our findings.

Regarding Review Comments to the Author:

Reviewer #1: - In this manuscript, Zafeiropoulou et al tried to investigate many aspects and phenotypes of the proteosome inhibitor (Bortezomib) resistant prostate cancer cell line. Despite presenting some preliminary interesting insights, there were many major concerns in the study:

1-Many of the outcomes of this study are a phenotype characterization of the prostate cancer bortezomib resistant cell line and not truly addressed the mechanistic pathways behind these resistance phenotypes as claimed by the aim of work and title of manuscript. The findings were a kind of association with Bortezomib resistance and not proved to be causative. The authors did not try to reverse the dysregulated markers observed in their results to see the corresponding effect on resistance and hence can build a causative relation.

Our findings are indeed a group of tightly related phenotypic observations in a Bortezomib-resistant clone we created in order to gather data on differences between resistant and non-resistant cells. The aim of our study was to create a model and observe its gradual transformation into a proteasome inhibitor cell line. In the meantime, we assessed the main cell functions to test whether the initial shock caused by Bortezomib had been overcome, and when the resistant clone emerged, we focused on the key points of the newly established phenotype. The differences we reported are thought to be indices of the underlying mechanisms and may become targets for future studies. Our study has a comparative idea on its grounds; we tried to simulate the long-term effects of Bortezomib treatment in prostate cancer cells and gather evidence on the alternations that emerged in the resistant clones. We acknowledge that neither we managed to detect the sole cause that drives Bortezomib resistance, nor did we manage to somehow reverse it; however, collecting a sum of findings and disseminating them to other research groups we thought would significantly accelerate the study of this phenomenon.

2- The novelty of the study is limited as it is very well established that resistant cancer cells in general including previous studies on bortezomib resistance in hematological cancers have increased proliferation, decreased apoptosis, increased wound healing efficiency, and alteration of cell cycle regulation and other essential kinases investigated here by authors.

The effects of Bortezomib in hematological cancers are long established; however, there is limited data on why the drug is ineffective in solid tumors as well as some types of hematological cancer like myelodysplastic syndromes. Bortezomib is a promising agent, as it is a mild chemotherapeutic, and exhibits low toxicity. Prostate cancer and other malignancies remain candidates for treatment with Bortezomib, but the drug’s low effectiveness limits its application. Clinical trials on prostate cancer have been conducted to some extent (Dreicer et al., 2007; Papandreou et al., 2004; Papandreou & Logothetis, 2004b); however, a lack of in-vitro data concerning resistance set the whole endeavor back. Given the established susceptibility of DU-145 to Bortezomib, we tried to indicate whether a resistant clone emerged after long-term drug administration, would possess characteristics already found in forms of cancer that are insusceptible to Bortezomib in the first place. These types of findings could explain whether some types of cancer (including prostate cancer) inherently possess alterations that make them insusceptible to the drug or whether they rapidly develop a phenotype similar to that developed in our model. Additionally, regarding the mechanisms studied, autophagy and oxidative stress are only recent advances in Bortezomib-resistance research (Cui et al., 2018; Kuczler et al., 2021; Quinet et al., 2022); therefore, assaying our cell model for these parameters offers new insight on how the UPS is substituted or regulated.

3- Point mutations or overexpression of β5 subunit of proteasome is a well-established cause of bortezomib resistance. This was also replicated by authors here with gene expression analysis showing overexpression of β5 subunit in the resistant cell line compared to control cell line. (a) However, the authors did not add another novel mechanistic layer explaining the reason behind this increased expression (like genetic or epigenetic factors). (b) It was not addressed by the authors whether this was the sole cause of resistance in their resistant cell line and then the other observed results are simply secondary consequences of this overexpression or there are additional novel mechanisms of resistance independent from β5 subunit overexpression. (c) Knockdown of PSMB5 may be helpful to see if this can reverse the resistance completely or partially. (d) In addition, sequencing of PSMB5 gene was also essential to investigate any point mutations which can cause hyperactive variant or hinder the binding with Bortezomib.

(a) The overexpression of PSMB5 is a well-established resistance mechanism in Multiple Myeloma (Oerlemans et al., 2008); however, no similar studies have been conducted to verify the very same mechanism in prostate cancer cells. Additionally, other groups have shown that PSMB5 expression in DU-145 cells is controlled by STAT3 activation (Vangala et al., 2014), and in our clone, we showed that STAT3 is overexpressed. The activation of the JAK/STAT pathway was found to be active in our clone, alongside members of the MAPKs and AKT pathways that are able to activate STATs, and therefore this was theorized as adequate by our team as an observation to partly explain the PSMB5 overexpression. It is of paramount importance for additional studies to be conducted in the field to improve our understanding of these observations.

(b) Quinet et al. in 2022 showed that Bortezomib-resistant mantle cell lymphoma cells relied on autophagy to substitute for the inactivated proteasomes (Quinet et al., 2022). The p62 marker was found to be overexpressed in their cells, an observation also made by our team for the first time in prostate cancer cells. Therefore, we proposed the activation of autophagy in the resistant clones as a crucial resistance mechanism alongside the cell’s induction of PSMB5 expression. Also, as mentioned by Chun KS, Jang JH, and Kim DH in 2020 “the redox modulation of critical cysteine residues present in the DNA-binding domain of STAT3 inhibits its DNA-binding activity” (Chun et al., 2020), and in our resistant model, we observed diminished oxidative stress levels in the resistant cells, especially in the absence of Bortezomib. These results indicated that the resistant cells have upregulated their antioxidant defense mechanisms, exploited also as a way to increase STAT signaling and induce the expression of proliferation- and metabolism-related genes (among which lies the PSMB5 gene). All these observations, combined, offer important insight into prostate cancer Bortezomib resistance, introducing a sum of characteristics as a manifold resistance phenotype.

(c) Studies knocking out this specific gene in TNBC (Wei et al., 2018)and in the MDA-MB-231 cell line (Wang et al., 2017) have shown inhibition of cell growth and migration as well as apoptosis induction. Additionally, knocking down the PSMB5 gene is an already proven way to resensitize cells to Bortezomib and thus cancel the acquired resistance in multiple myeloma cell lines (Shi et al., 2020). However, our intention was not solely to investigate whether a mutation or not facilitates Bortezomib resistance but rather to elucidate a wide panel of alterations observed phenotypically in resistant cells that distinguish them from non-resistant clones as a more holistic approach to resistance emergence.

(d) This study focused mainly on the signal transduction level rather than identifying mutations that emerged in the cell line. The PSMB5 overexpression documented in the resistant cells was theorized as the causal link between the measured higher ChT-L and the elevated PSMB5 protein levels. Alterations caused by point mutations in the molecule’s structure could alter Bortezomib’s binding ability or ChT-L activity. However, inconsistencies between PSMB5 quantity and activity, were not observed. Mutations of the PSMB5 gene have already been documented in (acquired Bortezomib-resistant) multiple myeloma cell lines (Oerlemans et al., 2008; Ri et al., 2010); however, we did not intend to identify possible mutations that emerged and accumulated in our clones rather than identify phenotypical differences that could emerge simultaneously with resistance in our DU-145 cell line that could resemble characteristics of inherent Bortezomib-tolerant solid tumors and hematological malignancies. We truly believe that the best application or confirmation of our study would be an analogous characterization of primary cultures of human prostate cancer cells, where gene sequencing will provide a better, in vivo perspective. Up to this day, no such study has been conducted; therefore, the real, in-vivo data are limited. Additionally, given the extended period of time that our cell model was exposed to Bortezomib, a condition that no patient would have undergone since Bortezomib is administered for short intervals, it made us think that sequencing the PSMB5 gene in our cell line would not offer insight into the mechanisms we were investigating. However, given the increased interest on Bortezomib resistance and the accessibility to sequencing technologies, we truly believe that future studies on the field, should extend from the phenotypic level we assessed and to the genotypic characterization.

4- The authors did not discuss the rationale of using prostate cancer cells especially that bortezomib is used commonly in cancers of hematopoietic origin and will be more clinically relevant in hematological cell lines.

It was indeed a mistake by us not to emphasize the use of prostate cancer cells in this study, and we are glad you helped us give prominence to it by changing our paper title and, in some cases, content. Although Bortezomib is commonly used in cancers of hematopoietic origin, limited data exist in solid tumors, including our own model, prostate cancer. Only a few clinical trials have been conducted on cancers of non-hematopoietic origin since many difficulties have to be overcome, concerning Bortezomib’s low specificity and permeability as well as the reduced bioavailability (Liu et al., 2022; Papandreou & Logothetis, 2004a). However, recent advances in biotechnology and drug delivery systems may provide novel, more efficient ways to deliver it to formerly inaccessible tumors and therefore expand its therapeutic benefits. For this to be feasible, in-vitro data shall be gathered by studying parameters like resistance and ineffectiveness and investigating the underlying causes. Recently, a new clinical trial was approved by the NIH in September (ClinicalTrials.gov Identifier: NCT06029998), to investigate the administration of Bortezomib in patients with Castration-Resistant Prostate Cancer, thus renewing interest in the drug. In addition, collecting data on the emerging phenotypes of various cancer types, may help to correlate them to hematological malignancies that show a diminished response to Bortezomib, such Myelodysplastic Syndromes. Overall, our data come to encourage more studies regarding the administration of Bortezomib and may act as a motive to further study the role of the pathways monitored in our paper in the regulation of ubiquitin-mediated proteolysis and its significance in disease.

5- The authors did not test their resistant cancer cell line against anti-cancer drugs from another family (something other than proteosome inhibitors) to see whether these observed phenotypes are really specific to Bortezomib or it is more of a cancer cell resistance to treatment in general.

This comment of yours was an idea we were able to embed in our paper since it was already being studied by our laboratory at the time of the peer review process. We tested the cells’ response to the anthracycline-class drug Doxorubicin, and we incorporated our findings in the revised version of the manuscript. Our findings verified that the cell clones we were studying had only acquired resistance to proteasome inhibitors (mainly Bortezomib, but to a lesser point Carfilzomib as well), regardless of the extended period of time that were cultured in a Bortezomib-rich environment that could trigger (more general) response mechanisms to drug-induced stress.

6-Title should be informative for the main message and outcomes (most significant result) of the study and not general description of the aim of study as authors do here.

We are delighted with your suggestion, as we now believe that the revised title better describes the outcomes of our study. Even though we investigated a wide spectrum of parameters, oxidative stress and autophagy are two cell functions designated as the most important findings of our paper. Therefore, we truly hope our revised title will give a more concise perspective of our work and emphasize our most important observations.

7- The figures are so redundant and can be merged at many occasions in one figure with many panels and some others can be transferred as supplementary figures.

We completely followed your suggestion in the figure issue, and we compiled a more dense and pluralistic way to present our results. Additionally, we added information in our results section and figure legends to better present the most significant outcomes, and we transferred to the appendix data that could be better represented with tables or charts, rather than images. 

8- Western blots are lacking the essential loading controls (figure 5A, 7, 9,10 and 11)

Loading controls are not presented because we did not use housekeeping proteins as reference genes; rather, we used the Bradford Assay to ensure the analysis of equal protein quantities. Given the increased cell size of resistant cells (visible in Figure 5B (revised version)), we did not trust the quantity of β-actin or tubulin as a reference protein. Additionally, treatment with Bortezomib which disrupts cell proteostasis, could lead to the accumulation of proteins that are thought to have a stable and uniform concentration among different cells. Therefore, we carefully ensured that equal numbers of cells were used to prepare the SDS-PAGE samples, and the desired protein quantity was determined using the Bradford Assay.

We would like to thank you again for the comments you made to our work, as we believe they really upgraded our manuscript. We also hope our revised version of the paper and our responses to your review comments will fully address your concerns and suggestions.

Sincerely,

Panagiotis Katsoris

Professor,

Department of Biology, University of Patras

Response to Reviewer 2:

Dear Dr. Gad,

Thank you for your review of our paper. Your suggestion truly helped us improve the Discussion section of our manuscript as you will see in the revised version.

Regarding Review Comments to the Author:

Reviewer #2: - This study Investigated the mechanisms underlying Bortezomib resistance. The article is well-written. I have no critical comments, but I hope to see soon further studies in in vivo models either in human or experimental animals. Also, I wished that the authors discussed other signaling pathways that might be involved in drug resistance and the possibility of Bortezomib to affect these pathways.

We truly appreciate your comments on our work. We also believe that our findings shall be extended to in vivo models and primary cultures. Occasioned by your comments, we further discussed extensions of our findings in redox signaling and the regulation of metabolism. We tend to believe that Bortezomib resistance shall be studied at the proteome and metabolome levels, since this is a very powerful way to detect sums of changes that are tightly connected and co-regulated. Additionally, we added some new results regarding response to other chemotherapeutics (Doxorubicin) and we showed that the clone we created did not acquire multi-drug resistance but rather proteasome inhibitor specific resistance. 

We believe that Bortezomib resistance shall be further investigated, and we thank you again for your constructive comments. 

We hope our response and revised version fully addresses your comments.

Sincerely,

Panagiotis Katsoris

Professor,

Department of Biology, University of Patras

---

## [Decision Letter · Decision Letter 1]

29 Jan 2024

Autophagy and Oxidative Stress Modulation Mediate Bortezomib Resistance in Prostate Cancer

PONE-D-23-23776R1

Dear Dr. Katsoris,

We’re pleased to inform you that your manuscript has been judged scientifically suitable for publication and will be formally accepted for publication once it meets all outstanding technical requirements.

Kind regards,

Laila Adel Ziko

Academic Editor

PLOS ONE

Additional Editor Comments (optional):

Reviewers' comments:

Reviewer's Responses to Questions

**Comments to the Author**

1. If the authors have adequately addressed your comments raised in a previous round of review and you feel that this manuscript is now acceptable for publication, you may indicate that here to bypass the “Comments to the Author” section, enter your conflict of interest statement in the “Confidential to Editor” section, and submit your "Accept" recommendation.

Reviewer #2: All comments have been addressed

Reviewer #3: All comments have been addressed

2. Is the manuscript technically sound, and do the data support the conclusions?

Reviewer #2: Yes

Reviewer #3: Yes

3. Has the statistical analysis been performed appropriately and rigorously? 

Reviewer #2: Yes

Reviewer #3: (No Response)

4. Have the authors made all data underlying the findings in their manuscript fully available?

Reviewer #2: Yes

Reviewer #3: Yes

5. Is the manuscript presented in an intelligible fashion and written in standard English?

Reviewer #2: Yes

Reviewer #3: Yes

6. Review Comments to the Author

Reviewer #2: None, thanks. All my comments have been addressed and discussed by the authors. I believe the article is now valid for publication

Reviewer #3: The authors have done a good job of addressing all the concerns raised by reviewers. The article is well-structured, with clear presentation and interpretation of results. In conclusion, this comprehensive study significantly contributes to the understanding of Bortezomib resistance in prostate cancer cells. The detailed exploration of genetic, cellular, and molecular mechanisms provides a solid foundation for future research and potential therapeutic strategies to overcome resistance.

7. PLOS authors have the option to publish the peer review history of their article (what does this mean?). If published, this will include your full peer review and any attached files.

Reviewer #2: No

Reviewer #3: No

---

## [Editor Report · Acceptance letter]

17 Feb 2024

PONE-D-23-23776R1 

PLOS ONE

Dear Dr. Katsoris, 

I'm pleased to inform you that your manuscript has been deemed suitable for publication in PLOS ONE. Congratulations! Your manuscript is now being handed over to our production team.

Kind regards, 

on behalf of

Dr. Laila Adel Ziko 

Academic Editor

PLOS ONE